# Comparison of frailty instruments for predicting mortality and prolon ged hospitalization in acute coronary syndrome patients

Anne Langsted[1], Jocelyne Benatar[2], Andrew Kerr[3], Katherine Bloomfield[4], Gerry Devlin[5], Alexander Sasse[6], David Smythe[7], Andrew To[8], Harvey White[2], Gerrard Wilkins[9], Ralph Stewart[2]*

1 Department of Clinical Biochemistry, Rigshospitalet, Denmark, 2 Cardiology Department, Health New Zealand Te Whatu Ora Te Toki Tumai, Green Lane Cardiovascular Services, Auckland, New Zealand, 3 Department of Epidemiology and Biostatistics, University of Auckland, Auckland, New Zealand, 4 Older Peoples Health, Waitemata, Auckland, New Zealand, 5 Cardiology Department, Gisborne Hospital, Gisborne, New Zealand, 6 Cardiology Department, Wellington Hospital, Wellington, New Zealand, 7 Cardiology Department, Christchurch Hospital, Christchurch, New Zealand, 8 Cardiology Department, North Shore Hospital, Auckland, New Zealand, 9 Cardiology Department, Dunedin Hospital, Dunedin, New Zealand

* rstewart@adhb.govt.nz

**Data Availability Statement:** Governance of study data, which was collected as part of ANZACS-QI, is described in the reference by Kerr (20). To ensure National Ethics Committee guidelines, including

## Abstract

### Objective

To evaluate the relative strengths of 3 frailty assessment instruments for predicting mortality and prolonged hospitalization in acute coronary syndrome patients.

### Design

Prospective cohort study.

### Setting

Acute cardiac care units in New Zealand.

### Participants

1174 patients >70 years of age hospitalized with an acute coronary syndrome.

### Interventions

The Clinical Frailty Scale (CFS), Edmonton Frail Scale (EFS) and Fried Criteria (Fried), were completed during hospital admission following an acute coronary syndrome when the patient was clinically stable.

### Primary and secondary outcome measures

All-cause mortality over the next ~5 years and hospitalization for >10 days in the next year determined from national administrative data.

those related to the Treaty Of Whitangi are met, all analyses of study data must be approved by the Vascular Informatics using Epidemiology and the Web (VIEW) governance committee, University of Auckland, New Zealand. Requests related to data study can be made to the corresponding author, Professor Ralph Stewart, e-mail rstewart@adhb.govt.nz.

**Funding:** The study was funded by the Health Research Council of New Zealand (HRC 14/689). Anne Langsted was supported by the Research Fund for the Capital Region of Denmark (A7160), the Beckett Foundation (21-2-7785), the Augustinus Foundation (21-2056), and the Reinholdt W. Jorck and Wife Foundation (21-JU-0053).

**Competing interests:** The authors have declared no competing interests exist.

**Abbreviations:** CFS, Clinical frailty scale; CHD, Cornary heart disease; CI, Confidence Interval; EFS, Edmonton frail scale; Euroscore II., European system for cardiac surgery risk evlaution score, version 2; Freid, Freid frailty score; GRACE, Global Registry of Acute Coronary Events; IDI, Incremental Discrimination Improvement; OR, Odds ratio.

## Results

During median follow-up of 5.1 years there were 353 deaths. Harrell's C-statistic for mortality for EFS was 0.663, Fried 0.648 and CFS 0.640 (p<0.001 for all). C-statistics for hospitalization >10 days (n = 267, 22%) were EFS 0.649, Fried 0.628, and CFS 0.584 (p<0.001 for all). Associations between increase in frailty scores and mortality were graded including in patients not classified as frail. The hazard ratio (HR) for mortality, adjusted for age and sex, for patients with an EFS score ≥9 (n = 197) compared to ≤2 (n = 331) was 5.0 (95% CI: 3.4–7.4). In models which included the Euroscore II or GRACE risk scores the EFS improved risk discrimination for both mortality and prolonged hospitalization more than the CFS and Fried.

## Conclusion

In older patients assessed following an acute coronary syndrome the EFS discriminated the risk of all cause mortality and prolonged hospitalization better than the CFS and Fried tests, and improved risk discrimination when added to clinical risk scores.

## Background

Frailty is a clinical syndrome characterized by multi-system impairment that decreases physiologic reserve and increases the vulnerability to stress [1]. Frailty is known to be associated with lower quality of life, prolonged hospitalisations, and increased all cause mortality [2–4]. The risks relative to benefits of some treatments may be higher for frail patients [5]. Clinical practice guidelines therefore recommend that frailty is considered in treatment decisions for older patients with cardiovascular diseases, including acute coronary syndromes [6–9].

A number of frailty assessment instruments are currently used. These include the Clinical Frail Scale (CFS) [10], the Fried criteria (Fried) [11] and the Edmonton Frail Scale (EFS) [12] Many studies have demonstrated associations between frailty assessed using these and other instruments and adverse clinical outcomes [13–15]. However, fewer studies have directly compared instruments for discriminating the risk of adverse clinical outcomes. Differences between 'frailty' tools may also be important to how frailty is interpreted clinically. The aim of this prospective cohort study was to compare three standard frailty instruments for predicting all cause mortality and prolonged hospitalisation in older patients admitted to hospital with an acute coronary syndrome.

## Methods and materials

### Study population

Patients aged more than 70 years admitted to a participating acute cardiac unit in New Zealand with an initial diagnosis of acute coronary syndrome were eligible for inclusion. Patients of Māori and Pacific descent could be included if >60 years, because age-related risk factors are present at a younger age in these individuals. 1231 patients were included from 5 New Zealand hospitals between August 4, 2015 and August 29, 2017. 57 patients were excluded because frailty assessments were incomplete, so the study population evaluated was 1174 patients.

All patients provided informed consent. The study was approved by the National Health and Disciplinary Ethics Committee, and the Maori Research Review Committee.

## Frailty instruments

Frailty instruments assessed in this study were the Clinical Frail Scale (CFS) [10], the Fried criteria (Fried) [11], and the Edmonton Frail Scale (EFS) [12]. The 3 frailty instruments were chosen because they were in common use, are relatively simple to complete, and differ in the way frailty is assessed.

The CFS is based on data from the Canadian Study of Health and Aging and consists of 7 categories ranging from very fit to severely frail, with each category defined by a short description [10]. Because the CFS most closely reflects overall clinical judgment, this scale was chosen to describe clinical risk factors and scores for other instruments by increasing frailty.

The Fried was derived with data from the Cardiovascular Health Study and defined by five criteria: unintentional weight loss in the past year, self-reported exhaustion, weakness measured by grip strength, slow walking speed, and low physical activity [11].

The EFS was developed for non-trained staff to assess frailty and includes 10 areas with a maximum score of 17 for the frailest individuals. The areas tested are mood, functional independence, social support, use of medication, nutrition, health perception, quality of life, continence, and two functional tests: the Clock test and the Timed get Up and Go test [12].

Frailty assessments were undertaken by a trained research nurse >24 hours after admission and when the patient was clinically stable, after obtaining written informed consent. Response to questions considered the patients status during the 2 weeks before the acute illness leading to hospitalization. The CFS was completed by the clinical team caring for the patient. Results of assessments were available to clinicians caring for the patient and included in the medical record.

## Clinical risk scores

To compare incremental risk discrimination of frailty instruments in additional to established clinical risk markers, the GRACE 6 month risk score [16] and Euroscore II were evaluated [17]. The GRACE discharge to 6 month mortality risk (%) is estimated from 9 variables: age, prior history of myocardial infarction, history of heart failure, heart rate at presentation, systolic blood pressure at presentation, initial serum creatinine, elevated cardiac biomarker levels, ST-segment depression on presenting electrocardiogram, and not having a percutaneous coronary intervention performed in hospital [16].

The Euroscore II estimates 30 day mortality after cardiac surgery [17]. Euroscore has also been associated with mortality in coronary heart disease patients treated by percutaneous coronary intervention [18] and with mortality during ten year follow up [19]. The Euroscore II includes points for New York Heart Association dyspnea grade, insulin-dependent diabetes mellitus, extracardiac arteriopathy, chronic lung disease, pulmonary hypertension, poor mobility, previous cardiac surgery, renal dysfunction, left ventricular ejection fraction, additional major procedures needed (eg valve or aortic root replacement), and recent myocardial infarction [19]. The study did not include patients with ongoing Class 4 angina, active endocarditis, critical pre-operative state, or need for emergency cardiac surgery.

## Linkage to registry and administrative data

The All of New Zealand, Acute Coronary Syndrome–Quality Improvement (ANZACS-QI) registry [20] has been completed in all cardiac centers and catheterization laboratories in New Zealand since 2015[20]. The ANZACS-QI is a web-based system designed to provide information on the quality of care of patients hospitalised with an acute coronary syndrome. Data completed before hospital discharge include age, sex, ethnicity, acute coronary syndrome diagnosis, current smoking, history of diabetes, hypertension requiring pharmacotherapy, prior

cardiovascular disease, history of heart failure, Grace 6 month risk score, and percutaneous coronary intervention or referral for coronary artery bypass surgery. Frailty assessments were recorded in a research module linked to the ANZACS-QI registry.

All individuals in New Zealand in contact with the health system are assigned a unique number, the National Health Index which can be linked to national health databases. The National Minimum dataset was used to identify all hospitalisations and deaths. The pre-specified primary endpoints of the current study were all-cause mortality from the National Mortality dataset up to September 2021 and hospitalization for any reason for more than 10 days during 12 months after the index admission. Outcomes were known for all 1174 study participants.

## Statistical analysis

Baseline clinical characteristics and summary statistics for the different instruments are presented according to the 7 categories of increasing frailty from the CFS. Values were reported as means and medians with inter-quartile range. For other analyses responses on each of the instruments are reported grouped using previously reported thresholds for 'frailty', or in approximate quintiles of the study population.

The associations between the frailty scores for each instrument and risk of death were analyzed using Cox proportional regression models and reported by hazard ratios with 95% confidence intervals adjusted by age and sex. For the Cox regressions, follow-up time started at the day of examination and ended at the day of death, or end of follow-up, whichever came first, using left truncation at study entry and age as underlying time scale. The associations between frailty and hospitalization >10 days in the next year were analysed using logistic regression, as the date of admission was less relevant.

The performance of different tools were evaluated by Harrell's C statistics and the integrative discriminative index (IDI). The C-statistics were evaluated adjusted for age and sex and in further analyses also adjusted for GRACE 6 month risk score and Euroscore II. IDI is the improvement in the difference in average predicted risks between the individuals with and without the outcome in the updated model, thereby the average improvement in sensitivity across all cutoffs [21]. All statistical analyses were done by STATA version 15.1. IDI for mortality and prolonged hospitalisation are compared for each frailty instrument when added to the GRACE 6 month mortality risk, Euroscore II, and the Clinical Frail Scale, as well as age and sex. Secondary analysis evaluated associations stratified for patients age <80 and ≥80 years.

## Data availability statement

Governance of study data, which was collected as part of ANZACS-QI, is described in the reference by Kerr [20]. To ensure National Ethics Committee guidelines, including those related to the Treaty Of Whitangi are met, all analyses of study data must be approved by the Vascular Informatics using Epidemiology and the Web (VIEW) governance committee, University ot Auckland, New Zealand. Requests related to data study can be made to the Co-chair of the ANZACS-QI Dr William Harrison, e-mail Wil.Harrison@middlemore.co.nz.

## Results

The mean age of the 1174 study participants was 76 years (inter-quartile range: 72–80), 37% were women, and 85% were of New Zealand or other European descent, 6% New Zealand Maori, 6% Pacific Islanders, and 3% of Asian descent. The discharge diagnosis was ST elevation myocardial infarction in 23%, non-ST elevation myocardial infraction in 71%, and not an acute coronary syndrome in 6%.

**Table 1. Clinical characteristics of study population reported according to the clinical frail scale.**

| | | Clinical Frail Scale | | | | | | |
|---|---|---|---|---|---|---|---|---|
| | All | Very fit | Well | Well comorbid | Vulnerable | Mildly frail | Moderately frail | Severely frail |
| **Number** (%) | 1174 (100) | 113 (10) | 182 (16) | 333 (28) | 280 (24) | 158 (13) | 99 (8) | 9 (1) |
| Age | 76 (72–80) | 76 (73–80) | 75 (71–79) | 76 (72–79) | 76 (73–80) | 78 (73–83) | 77 (74–83) | 75 (74–83) |
| Women, % | 37 | 23 | 39 | 34 | 34 | 53 | 47 | 22 |
| **Ethnicity**, N(%) | | | | | | | | |
| NZ European | 810 (70) | 80 (72) | 120 (66) | 239 (72) | 187 (67) | 110 (71) | 68 (72) | 6 (67) |
| Other European | 174 (15) | 23 (21) | 31 (17) | 38 (12) | 49 (18) | 20 (13) | 11 (12) | 2 (22) |
| NZ Maori | 73 (6) | 6 (5) | 7 (4) | 27 (8) | 15 (5) | 10 (6) | 7 (7) | 1 (11) |
| Pacific Islands | 65 (6) | 0 (0) | 14 (8) | 14 (4) | 23 (8) | 10 (6) | 4 (4) | 0 (0) |
| Asian | 38 (3) | 2 (2) | 9 (5) | 12 (4) | 5 (2) | 6 (4) | 4 (4) | 0 (0) |
| **CVD risk factors** | | | | | | | | |
| Current smoker | 74 (7.3) | 2 (2.1) | 7 (4.3) | 28 (9.8) | 23 (9.5) | 8 (6.1) | 6 (7.2) | 0 (0) |
| Diabetes | 292 (29) | 20 (21) | 27 (17) | 80 (28) | 84 (35) | 42 (32) | 35 (42) | 4 (50) |
| BMI ≥30 | 254 (31) | 13 (17) | 42 (33) | 77 (32) | 73 (38) | 25 (24) | 22 (34) | 2 (29) |
| BMI ≤20 | 25 (3.1) | 2 (2.6) | 4 (3.2) | 2 (0.8) | 5 (2.6) | 7 (6.7) | 5 (7.7) | 0 (0) |
| Prior CVD | 452 (39) | 36 (32) | 56 (31) | 133 (40) | 129 (46) | 56 (35) | 38 (38) | 4 (44) |
| Prior MI | 285 (24) | 24 (21) | 30 (16) | 79 (24) | 86 (31) | 35 (22) | 29 (29) | 2 (22) |
| Heart failure | 62(5) | 2(1.8) | 3(1.7) | 18(5.4) | 24(8.6) | 7(4.4) | 8(8.1) | 0(0) |
| COPD | 121(10) | 4(3.5) | 12(6.6) | 30(9.0) | 31(11) | 22(14) | 19(19) | 3(33) |
| Hemoglobin, mg/dl | 135(124–147) | 142(131–153) | 138(128–149) | 136(126–149) | 135(124–144) | 130(120–139) | 126(113–138) | 125(107–138) |
| Creatinine, mmol/l | 94 (80–132) | 94 (79–105) | 90 (77–104) | 93 (80–111) | 97 (81–118) | 97 (78–129) | 103 (84–128) | 108 (93–132) |
| **Discharge diagnosis** | | | | | | | | |
| STEMI | 233 (23) | 24 (25) | 35 (22) | 66 (23) | 55 (23) | 29 (22) | 20 (24) | 4 (50) |
| Non-STEMI | 716 (71) | 68 (71) | 111 (68) | 214 (75) | 169 (70) | 91 (69) | 59 (71) | 4 (50) |
| Not ACS | 61 (6) | 4 (4) | 17 (10) | 7 (2) | 17 (7) | 12 (9) | 4 (5) | 0 (0) |
| **Coronary revascularisation** | | | | | | | | |
| PCI | 467 (40) | 50 (44) | 68 (37) | 142 (43) | 111 (40) | 55 (35) | 39 (39) | 2 (22) |
| CABG | 185 (16) | 25 (27) | 44 (24) | 56 (17) | 38 (14) | 14 (9) | 8 (8) | 0 (0) |
| No coronary revascularisation | 522 (44) | 38 (39) | 70 (39) | 135 (40) | 131 (46) | 89 (56) | 52 (53) | 7 (88) |

Numbers are median (interquartile range) or number (percent). N = number. NZ = New Zealand.

ACS = Acute coronary syndrome, BMI = Body mass index in kg/m², CABG = coronary artery bypass graft surgery, COPD = chronic obstructive pulmonary disease, CVD = cardiovascular disease, MI = myocardial infarction, PCI = percutaneous coronary intervention, STEMI = ST elevation myocardial infarction.

The CFS classified 10% of study participants as very fit, 16% well, 28% well with comorbidities, 24% vulnerable, 13% mildly frail, 8% moderately frail, and 1% severely frail. Baseline characteristics of the study population are displayed in Table 1 across 7 categories of the CFS. Increasing CFS score was associated with an increasing percentage of patients with diabetes, lower hemoglobin and higher creatinine levels. Frail patients were less likely to undergo percutaneous coronary intervention or coronary artery bypass surgery. Increasing CFS grade was also associated a higher Euroscore II and GRACE score, and with greater frailty assessed using the EFS and Fried criteria (Table 2).

## Frailty scores and outcome

During a median follow-up of 5.1 (Interquartile range (IQR): 4.6–5.5) years there were 353 deaths from all causes (29%). During 12 months after the index hospitalization 267 patients (22%) were hospitalized for > 10 days. C-statistics for mortality and hospitalization for > 10 days are compared for the different tools in Table 3. The discrimination of all-cause mortality according to Harrell's C-index for EFS was 0.663 (95%CI: 0.635–0.692), Fried 0.648 (0.614–0.683), and the CFS 0.641 (0.610–0.671). The C-statistics for hospitalization >10 days in the

**Table 2. Scores for the Edmonton Frail Scale (EFS), Fried score and the GRACE and Euroscore II clinical risk scores for the study population reported by increasing level of frailty according to the clinical frail scale.**

|  |  |  | Clinical Frail Scale |  |  |  |  |  |
|---|---|---|---|---|---|---|---|---|
|  | **All** | **Very fit** | **Well** | **Well comorbid** | **Vulnerable** | **Mildly frail** | **Moderately frail** | **Severely frail** |
| **Number (%)** | 1174 (100) | 113 (10) | 182 (16) | 333 (28) | 280 (24) | 158 (13) | 99 (8) | 9 (1) |
| **Frailty assessment** |  |  |  |  |  |  |  |  |
| **EFS** | 4 (2–7) | 2 (1–4) | 3 (2–5) | 4 (2–6) | 5 (3–7) | 7 (4–9) | 9 (6–11) | 12 (10–13) |
| **Fried** | 1 (1–2) | 1 (0–2) | 1 (0–2) | 1 (0–2) | 2 (1–2) | 3 (1–3) | 3 (2–4) | 4 (4–5) |
| **Clinical risk scores** |  |  |  |  |  |  |  |  |
| **GRACE (% risk)** | 3 (1–6) | 3 (1–4.5) | 2 (1–4) | 3 (1–5) | 3 (1–7) | 3 (1.5–6) | 4 (2–7) | 17 (3–24) |
| **Euroscore II (% risk)** | 2 (2–4) | 2 (1–3) | 2 (1–3) | 2 (1–4) | 3 (2–5) | 3 (2–6) | 4 (2–6) | 10 (4–13) |

Numbers are mean (interquartile range) for each group.

EFS = Edmonton Frail Scale, GRACE = Global Registry of Acute Coronary Events 6 month risk score. EuroSCORE II = The European System for Cardiac Operative Risk Evaluation Score II

next year were higher for EFS 0.649 (0.611–0.687) and the Fried 0.628 (0.585–0.672) compared to the CFS 0.566 (0.523–0.609).

The HR for all cause mortality increased with increasing frailty for the EFS, CFS and Fried scores (Fig 1). Increasing frailty assessed using the CFS, EFS and Fried scores were each associated with a higher odds ratios (OR) for hospitalization lasting >10 days during the next year (Fig 2).

The graded association between increase in EFS score and mortality was also observed in patients with an EFS score ≤5, who do not meet criteria for being frail. Compared to low EFS scores (score 0–2, n = 331 of 1174, 28%), the hazard ratio (HR) for all-cause mortality for patients with higher scores (9–17, n = 197 of 1174, 17%) was 5.0 (95% confidence interval (CI): 3.4–7.4) (Fig 3).

Each of the frailty tools improved risk discrimination for both mortality and prolonged hospitalization when added to either the Euroscore II or the GRACE risk score. The incremental improvement in risk discrimination (IDI) was greater for the EFS compared to the CFS and Fried scores (Table 4). When combining information from different frailty insrtruments both EFS and Fried scores improved risk discrimination when added to the CFS (Table 4).

In a secondary analysis, discrimination for mortality is compared for the 871 patients aged <80 years with the 360 patients aged ≥80 years. For each of the 3 frailty scales C-statistics for mortality were higher for patients aged <80 compared to ≥80 years years, and higher for the EFS compared to the Fried and CFS. (Table 5, Fig 3).

## Discussion

### Associations between frailty scores and outcomes

In this study greater frailty was associated with higher all cause mortality and a higher risk of prolonged hospitalisation when assessed using three different frailty instruments. These frailty

**Table 3. Discrimination for mortality and for hospitalization of >10 days during the next year of different frailty assessment tools, adjusted for age, sex, Euroscore II and Grace risk scores.**

|  | Mortality |  | Hospitalization |  |
|---|---|---|---|---|
| **Frailty assessment tool** | **C-statistic** | **95% CI** | **C-statistic** | **95%CI** |
| Clinical Frail Scale | 0.619 | 0.580–0.659 | 0.566 | 0.523–0.609 |
| Edmonton Frailty Scale | 0.648 | 0.610–0.686 | 0.628 | 0.586–0.670 |
| Fried Criteria | 0.622 | 0.577–0.667 | 0.611 | 0.563–0.658 |

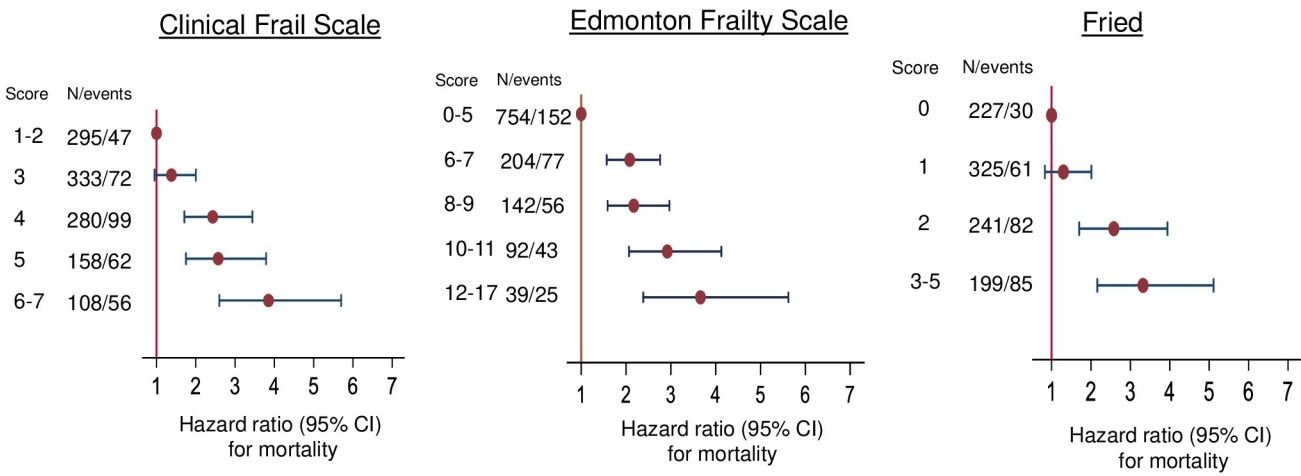

**Fig 1. Associatons between frailty scores using 3 instruments and mortality.** Hazard ratios with 95% confidence intervals for mortality by increasing frailty assessed using the clinical frail scale, the Edmonton Frail Scale and the Fried criteria for frailty. Hazard ratios are calculated using Cox regression analyses adjusted for age and sex. N = number of patients with frailty score / number who died 95%CI = 95%confidence interval.

instruments also added prognostic information to both the GRACE [16] and Euroscore II [17] scores which use clinical risk markers. In all evaluations which compared frailty instruments risk discrimination for mortality and prolonged hospitalization were stronger for the EFS when compared to the CFS and Fried scores.

The CFS [10], which was the simplest frailty instrument assessed in this study, provided incremental risk information when added to clinical risk scores. These observations support the conclusion that clinical assessment of frailty provides a reasonable assessment of prognosis. However risk discrimination was stronger for the EFS compared to the CFS, and the EFS improved risk discrimination further when added to the CFS. This suggests that frailty tools which use objective criteria can provide a more reliable assessment than those based on clinical judgment alone.

The EFS is a multidimensional assessment which includes evaluation of physical activity, weight loss, activities of daily living, cognitive function and social support [12]. Multi-

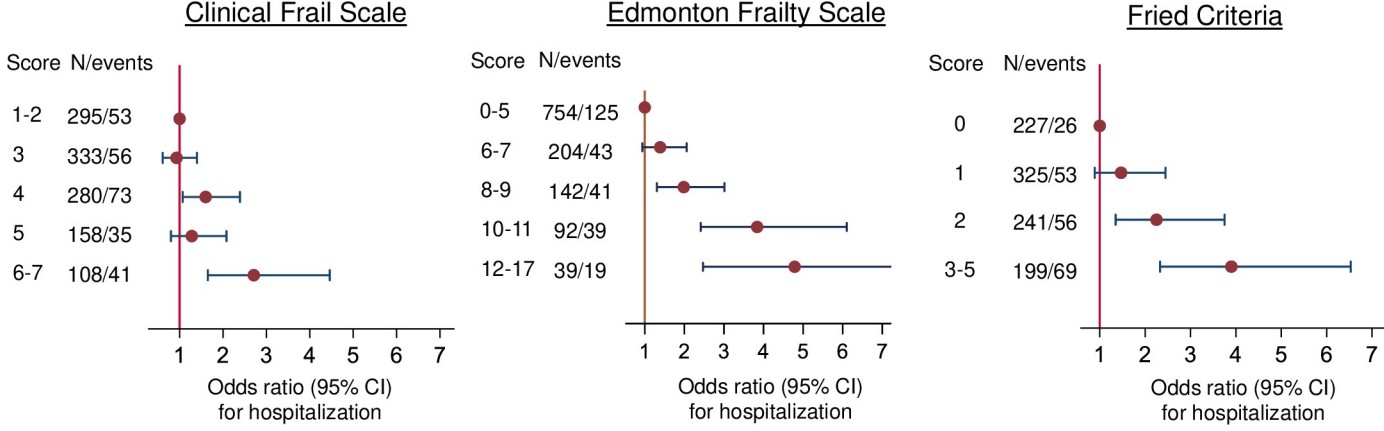

**Fig 2. Associations between frailty scores using 3 instruments and prolonged hospitalisation.** Odds ratios with 95% confidence intervals for hospitalisation for >10 days in the next year by increasing scores on the clinical frail scale, the Edmonton Frail Scale and the Fried criteria adjusted for age and sex. Score = score on each frailty tool, N = number of patients with score / number with prolonged hospitalisation. CI = confidence interval.

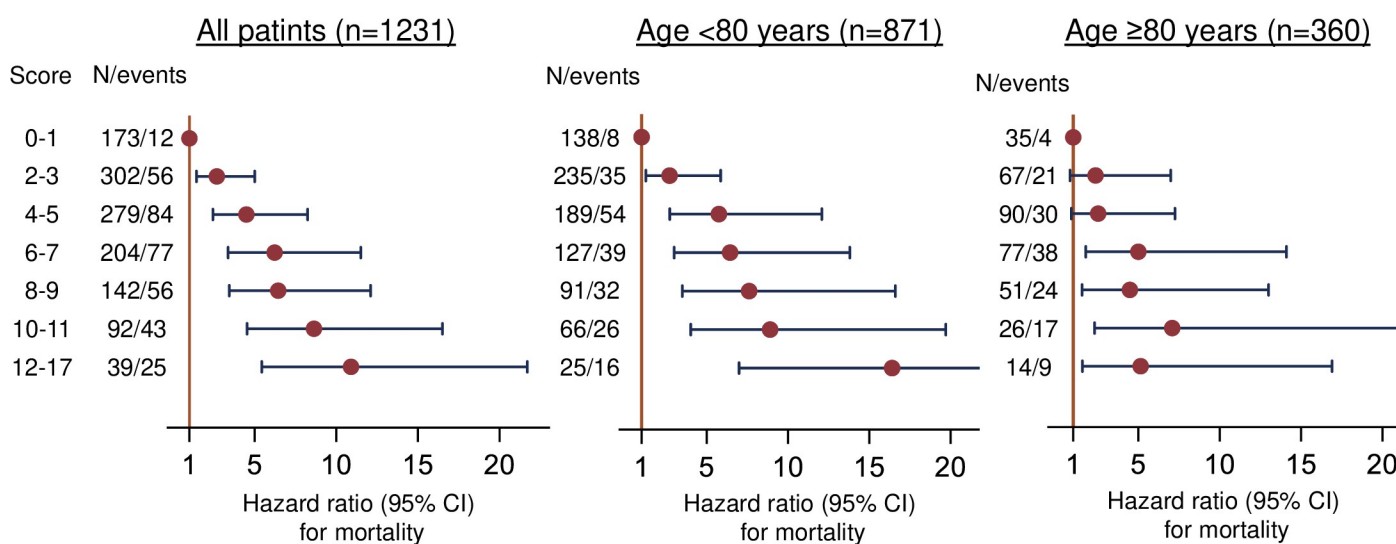

**Fig 3. Associations between Edmonton frailty scale score and mortality for all patients and patients aged <80 years and ≥80 years.** The EFS is reported for the range of scores including patients who do not meet criteria for mild or greater frailty (EFS score ≤5).

dimensional frailty assessments can identify reasons for greater 'frailty' which are relevant to clinical care. The multi-dimensional EFS also improved risk discrimination compared to the Fried score, which evaluates the 'frailty phenotype' with 5 questions related to physical function, strength and sacropenia [11]. In a study of patients hospitalized with heart failure a multi-dimensional frailty instrument also had better risk discrimination compared to a physical frailty score [29].

A strength of the EFS and CFS is the graded assessment of frailty. In this study 64% of patient did not meet the criteria for mild or greater frailty based on the EFS score. However differences in mortality rsk were graded both above and below standard thresholds for 'frailty'.

**Table 4. Integrated Discrimination Improvement (IDI) for predicting mortality and hospitalization >10 days when different frailty assessment tools are added to the clinical frail scale, the GRACE risk score, and Euroscore II risk score.**

|  | Clinical Frail Scale | GRACE score | Euroscore II |
|---|---|---|---|
| **IDI for mortality** |  |  |  |
| Clinical Frail Scale | - | 0.0550 | 0.0259 |
| Edmonton Frailty Scale | 0.0350 | 0.0744 | 0.0425 |
| Fried Criteria | 0.0282 | 0.0603 | 0.0280 |
| **IDI for prolonged hospitalisation** |  |  |  |
| Clinical Frail Scale | - | 0.0169 | 0.0090 |
| Edmonton Frailty Scale | 0.0387 | 0.0493 | 0.0409 |
| Fried Criteria | 0.0291 | 0.0396 | 0.0277 |

Higher Integrated Discrimination Improvement (IDI) indicates more improvement in risk discrimination. For all evaluations improvement in IDI was strongly statistically significant (p<0.0001).

IDI was greater when the EFS was added to the clinical frailty scale compared to the Fried score, and when compared to both the Fried score and CFS when added to the GRACE and Euroscore II risk scores.

**Table 5. Discrimination for mortality of 3 frailty tools stratified for acute coronary syndrome for 871 patients aged < 80 and 360 patients aged ≥80 years.**

| | <80 years | | ≥80 years | |
|---|---|---|---|---|
| Assessment tool | C-statistic | 95% CI | C-statistic | 95%CI |
| Clinical Frail Scale | 0.662 | 0.620–0.755 | 0.628 | 0.566–0.689 |
| Edmonton Frailty Scale | 0.692 | 0.651–0.733 | 0.645 | 0.586–0.704 |
| Fried score | 0.663 | 0.614–0.711 | 0.618 | 0.548–0.688 |

95%CI = 95% confidence interval.

## Comparison with other studies

Previous studies have reported associations between frailty assessed using the Fried criteria [22,23], the CFS [24–26], the EFS [27,28], as well as other frailty instruments [29] and all cause mortality and hospitalisations. However, few large studies have directly compared the utility of different frailty instruments in ACS patients. One study compared mortality and hospitalization risk for frail compared to non-frail patients assessed using the Fried, EFS, and CFS in 174 acute coronary syndrome patients [14]. However the study was too small to reliably evaluate differences between instruments.

## Clinical implications

Assessment of frailty may be relevant when considering cardiac interventions which carry a higher risk of adverse clinical outcomes, and when the benefits of intervention are modest compared to risks [13,30]. An example is the decision of coronary artery bypass surgery compared to medical therapy in patients with ischemic left ventricular dysfunction, in whom on average the benefits of surgery are less in older patients [31] and those with a lower exercise capacity [32]. In the current study coronary artery bypass surgery was performed in <10% of patients who were 'moderately frail' compared to 27% consider to be 'very fit', The Euroscore II is commonly used to estimate 30 day mortality risk after cardiac surgery [17], and also predicts mortality after percutaneously coronary intervention [18], and long term mortality [19]. The observation that each of the frailty instruments improved risk discrimination when added to Euroscore II supports including evaluation of frailty for patients considered for cardiac surgery.

A strength of the EFS is assessment of the level of frailty on a continuum between good health and advanced frailty. The level of frailty which would modify treatment decisions is likely to vary for different treatments and indications. A graded scale may also be more reliable for documenting changes over time.

In this study each of the frailty tools improved discrimination for mortality in patients aged <80 as much as for patients ≥80 years. Frailty assessments such as the EFS therefore improve risk discrimination in a broad range of older patients, including for patients who do not meet criteria for being 'frail'.

## Study limitations

The study population was representative of older patients with an acute coronary syndrome. However it did not include ACS patients not admitted to an acute cardiac care unit, because of advanced age, frailty or comorbidity. Results are likely to be applicable to patients with other cardiac diagnoses, but further research is need to confirm this. It is possible results could vary for different countries and ethnicities, and for patients with different medical problems. The study included diverse ethnic groups from within New Zealand, but was not designed to assess

ethnic differences. Further research is needed to compare frailty instruments in patients aged <70 years.

This study evaluated 3 widely used frailty tools, but did not evaluate all frailty instruments. Assessments undertaken during an acute medical admission may be less reliable indicators of usual status. Variation in frailty evaluations were not assessed, but likely to be similar to their use clinically.

Strengths of this study include use of standard frailty instruments as part of usual care in a large cohort of ACS patients, and follow-up from national administrative data with no missing outcome data.

## Conclusion

In acute coronary syndrome patients aged ≥70 years the Edmonton Frailty Scale improved discrimination for all-cause mortality more than the Clinical Frail Scale and Fried score, and improved risk discrimination when added to clinical risk scores.

## Author Contributions

**Conceptualization:** Anne Langsted, Ralph Stewart.

**Data curation:** Anne Langsted.

**Formal analysis:** Anne Langsted, Jocelyne Benatar.

**Investigation:** Ralph Stewart.

**Methodology:** Anne Langsted, Jocelyne Benatar.

**Project administration:** Ralph Stewart.

**Supervision:** Ralph Stewart.

**Writing – original draft:** Anne Langsted, Ralph Stewart.

**Writing – review & editing:** Anne Langsted, Jocelyne Benatar, Andrew Kerr, Katherine Bloomfield, Gerry Devlin, Alexander Sasse, David Smythe, Andrew To, Harvey White, Gerrard Wilkins, Ralph Stewart.

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
