## [Decision Letter · Decision Letter 0]

20 Aug 2024

PONE-D-24-28766Comparison of frailty instruments for predicting  mortality and prolonged hospitalization in acute coronary syndrome patientsPLOS ONE

Dear Dr. STEWART,

Thank you for submitting your manuscript to PLOS ONE. After careful consideration, we feel that it has merit but does not fully meet PLOS ONE’s publication criteria as it currently stands. Therefore, we invite you to submit a revised version of the manuscript that addresses the points raised during the review process.

We look forward to receiving your revised manuscript.

Kind regards,

Pasquale Abete

Academic Editor

PLOS ONE

Journal Requirements:

4. In the online submission form, you indicated that "The data is available on request and will be shared on reasonable requests to the corresponding author."

6. Please include a separate caption for each figure in your manuscript.

8. We note that there is identifying data in the Supporting Information file <file name>. Due to the inclusion of these potentially identifying data, we have removed this file from your file inventory. Prior to sharing human research participant data, authors should consult with an ethics committee to ensure data are shared in accordance with participant consent and all applicable local laws.

-Location data

Additional Editor Comments:

According to Reviewers' decision, the manuscript needs a major revision.

Reviewers' comments:

Reviewer's Responses to Questions

**Comments to the Author**

1. Is the manuscript technically sound, and do the data support the conclusions?

Reviewer #1: Partly

Reviewer #2: Yes

2. Has the statistical analysis been performed appropriately and rigorously? 

Reviewer #1: Yes

Reviewer #2: Yes

3. Have the authors made all data underlying the findings in their manuscript fully available?

Reviewer #1: Yes

Reviewer #2: Yes

4. Is the manuscript presented in an intelligible fashion and written in standard English?

Reviewer #1: Yes

Reviewer #2: Yes

5. Review Comments to the Author

Reviewer #1: In this prospective multicenter study of 1174 patients over 70 years old hospitalized with acute coronary syndrome, four frailty assessment tools and a cognitive function test were evaluated for their ability to predict mortality and prolonged hospitalization. All frailty assessment tools were associated with graded increases in mortality risk, with the Edmonton Frail Scale (EFS) achieving the highest c-statistic (0.663) compared to the others. The EFS also showed improved risk discrimination when added to the Euroscore II and GRACE clinical risk scores. For predicting prolonged hospitalization (>10 days), the EFS again had the highest c-statistic (0.649). The study concludes that the EFS is the most effective tool for discriminating the risk of mortality and prolonged hospitalization compared to other frailty assessments and the GPCog cognitive test.

The study is commendable for its focus on a highly relevant issue—assessing frailty in older patients with acute coronary syndrome. The clinical relevance is clear, as frailty is an important predictor of outcomes in elderly patients, and its accurate assessment can significantly influence management strategies. Additionally, the statistical methodology employed is robust and appropriate for the study design, ensuring reliable and valid results.

However, there are several points that warrant further consideration:

- From a methodological viewpoint, the inclusion of the GPCog cognitive test may not be necessary. Limiting the comparison to frailty assessment tools alone would make the study more streamlined and interpretable.

- The Katz index is primarily a measure of disability, which is a distinct phenomenon than frailty. The ADL levels are not highly discriminative in the studied population, and it is unclear whether the tables reference lost or maintained ADLs. So, it can be omitted from the analysis.

- In Figure 1, for the calculation of Cox regression for the EFS, I would like to see the following scoring: 0-5 = Not Frail; 6-7 = Vulnerable 8-9 = Mild Frailty; 10-11 = Moderate Frailty; 12-17 = Severe Frailty

- It should be noted that none of the frailty assessment tools used were specifically developed for acute settings, and there is a lack of evidence regarding their reproducibility in this context. This limitation should be explicitly mentioned.

- Although the methods state that the evaluations were conducted when patients were clinically stable, it is important to clarify whether the Clinical Frailty Scale refers to the level of frailty before the acute coronary syndrome event or at the time of the ACS. Given that the CFS reflects overall clinical judgment, there is a potential for overestimating frailty in patients assessed during this acute phase.

- In the discussion, it is essential to highlight the multidimensional contribution of frailty to cardiovascular diseases. Please refer to and discuss the findings of Testa et al. (2020), which address physical versus multidimensional frailty in older adults with and without heart failure: "Testa G, Curcio F, Liguori I, Basile C, Papillo M, Tocchetti CG, Galizia G, Della-Morte D, Gargiulo G, Cacciatore F, Bonaduce D, Abete P. Physical vs. multidimensional frailty in older adults with and without heart failure. ESC Heart Fail. 2020 Jun;7(3):1371-1380. doi: 10.1002/ehf2.12688. Epub 2020 Apr 3. PMID: 32243099; PMCID: PMC7261566."

Overall, this study makes a significant contribution to understanding frailty in elderly patients with acute coronary syndrome, and these considerations can further refine its impact and applicability.

Reviewer #2: The study evaluated the relative strengths of 4 frailty assessment instruments and a test for cognitive function for predicting mortality and prolonged hospitalization in acute coronary syndrome patients. 1174 patients >70 years of age hospitalized with an acute coronary syndrome were enrolled. All-cause mortality over the 5 years and hospitalization for >10 days in the next year was determined from national administrative data. Edmonton Frail Scale (EFS), Fried Criteria (Fried), Clinical Frailty Scale (CFS), Katz score (Katz), a dementia screening test (GPCog), the GRACE and Euroscore2 clinical risk scores were completed during hospital admission. During a median follow-up of 5.1 years there were 353 deaths. Frailty assessed using all instruments was associated with graded increases in mortality risk. Harrell’s C-statistic for mortality were EFS was 0.663, Fried 0.648, CFS 0.640, Katz 0.593, and GPCog 0.608 (p<0.001 for all). For patients with an EFS score 9 (n=197) compared to 2 (n=331) the hazard ratio (HR) for mortality, adjusted for age and sex, was 5.0 (95% CI: 3.4-7.4). C-statistics for hospitalization >10 days (n=267, 22%) were EFS 0.649, Fried 0.628, Katz 0.602, CFS 0.584, and GPCog 0.552, (p<0.001 for all). In models which included the Euroscore II and GRACE risk scores the EFS improved incremental risk discrimination for both mortality and prolonged hospitalization more than other frailty instruments.

The topic of the study is of interest. I have some suggestions. ADL are measure of disability and I think not appropriate to use Kats index as measure of frailty, similarly GPCog measure one of the frailty domain. The AUC found in your analysis could also be evaluating stratifying the sample by age (i.e. 70-80 and > 80). It should be of interest to evaluate the results of a cox regression analysis on your outcome to verify the effect of the different instruments of frailty. Why measure in this setting Euroscore II?

6. PLOS authors have the option to publish the peer review history of their article (what does this mean?). If published, this will include your full peer review and any attached files.

Reviewer #1: No

Reviewer #2: **Yes: **Francesco Cacciatore

---

## [Author Response · Author response to Decision Letter 0]

12 Nov 2024

Reviewer #1: 

- From a methodological viewpoint, the inclusion of the GPCog cognitive test may not be necessary. Limiting the comparison to frailty assessment tools alone would make the study more streamlined and interpretable.

- The Katz index is primarily a measure of disability, which is a distinct phenomenon than frailty. The ADL levels are not highly discriminative in the studied population, and it is unclear whether the tables reference lost or maintained ADLs. So, it can be omitted from the analysis.

Reply: The GP Cog and Katz scores are now not included.

Reviewer #1: 

In Figure 1, for the calculation of Cox regression for the EFS, I would like to see the following scoring: 0-5 = Not Frail; 6-7 = Vulnerable 8-9 = Mild Frailty; 10-11 = Moderate Frailty; 12-17 = Severe Frailty.

Reply: The scoring of frailty using the EFS has been changed in figure 1 as suggested, and revised the methods and results sections accordingly.

An additional observation was a graded increase in mortality and prolonged hospitalisation risk with increase for patients with an EFS score <=5, who do not meet these criteria for frailty. We have illustrated this in an additional figure 3, which also displays results stratified by age <80 and >=80, as suggested by reviewer 2. 

Reviewer #1: 

It should be noted that none of the frailty assessment tools used were specifically developed for acute settings, and there is a lack of evidence regarding their reproducibility in this context. This limitation should be explicitly mentioned.

Reply: 

We agree with this limitation. 

See discussion, study limitations. “Assessments undertaken during an acute medical admission may have been less reliable indicators of usual status.” 

Reviewer #1: 

Although the methods state that the evaluations were conducted when patients were clinically stable, it is important to clarify whether the Clinical Frailty Scale refers to the level of frailty before the acute coronary syndrome event or at the time of the ACS. Given that the CFS reflects overall clinical judgment, there is a potential for overestimating frailty in patients assessed during this acute phase.

Reply: See Methods section under ‘frailty instruments’ and limiyations section as above

 “Frailty assessments were undertaken by a trained research nurse >24 hours after admission and when the patient was clinically stable, after obtaining written informed consent. Response to questions considered the patient’s status during the 2 weeks before the acute illness leading to hospitalization…” 

Reviewer #1: 

In the discussion, it is essential to highlight the multidimensional contribution of frailty to cardiovascular diseases. Please refer to and discuss the findings of Testa et al. (2020), which address physical versus multidimensional frailty in older adults with and without heart failure: "Testa G, Curcio F, Liguori I, Basile C, Papillo M, Tocchetti CG, Galizia G, Della-Morte D, Gargiulo G, Cacciatore F, Bonaduce D, Abete P. Physical vs. multidimensional frailty in older adults with and without heart failure. ESC Heart Fail. 2020 Jun;7(3):1371-1380. doi: 10.1002/ehf2.12688. Epub 2020 Apr 3. PMID: 32243099; PMCID: PMC7261566."

Reply: Reference to the study by Testa (reference 29) has been included. See discussion section 

Under associations between frailty scores and outcomes.

“The EFS is a multidimensional assessment which includes evaluation of physical activity, weight loss, activities of daily living, cognitive function and social support (12). ……….. In a study of patients hospitalized with heart failure a multi-dimensional frailty instrument also had better risk discrimination compared to a physical frailty score (29) “ 

And Under comparison with other studies

“Previous studies have reported associations between frailty assessed using the Fried criteria (22, 23),, the CFS (24, 25, 26), the EFS (27, 28), as well as other frailty instruments (29) and all cause mortality and hospitalisations. 

Reviewer #2: 

I have some suggestions. ADL are measure of disability and I think not appropriate to use Kats index as measure of frailty, similarly GPCog measure one of the frailty domain. 

Reply: 

We agree the Katz and GP Cog are not measures of frailty, and have removed these from the manuscript.

Reviewer #2: 

The AUC found in your analysis could also be evaluating stratifying the sample by age (i.e. 70-80 and > 80). It should be of interest to evaluate the results of a cox regression analysis on your outcome to verify the effect of the different instruments of frailty. 

Reply:

We have undertaken an additional analysis stratified by age <80 years and >= 80 years as suggested. C-statistics are reported in table 5. This analysis demonstrated that comparisons of frailty tools were consistent across age groups, and that risk discrimination appears to be better for patients aged <80 years. Results of Cox regression analyses are displayed for EFS in the new figure 3.

Reviewer #2: 

Why measure in this setting Euroscore II?

Reply: 

The reasons for including adjustment for Euroscore II are now more clearly documented.

See methods section 

“The Euroscore II estimates 30 day mortality after cardiac surgery (17). Euroscore has also been associated with mortality in coronary heart disease patients treated by percutaneous coronary stenting (18) and with mortality during long ten year follow up (19). ……”

And discussion under ‘clinical implications’

“ ….. For example, in patients with ischemic left ventricular dysfunction the benefits of coronary artery bypass surgery compared to medical therapy are less in older patients (31) and those with a lower exercise capacity (32). In the current study coronary artery bypass surgery was performed in <10% of patients who were ‘moderately frailty’ compared to 27% graded as ‘very fit’, The Euroscore is commonly used to estimate 30 day mortality risk after cardiac surgery (19), and predicts mortality after percutaneously coronary intervention (18), and long term mortality (19), The observation that each of the frailty instruments improved risk discrimination when added to Euroscore II supports including evaluation of frailty for patients considered for cardiac surgery.

---

## [Decision Letter · Decision Letter 1]

21 Jan 2025

Comparison of frailty instruments for predicting  mortality and prolonged hospitalization in acute coronary syndrome patients

PONE-D-24-28766R1

Dear Dr. Stewart,

We’re pleased to inform you that your manuscript has been judged scientifically suitable for publication and will be formally accepted for publication once it meets all outstanding technical requirements.

Kind regards,

Barry Kweh

Academic Editor

PLOS ONE

Additional Editor Comments (optional):

The authors have clarified methodological queries, refined as well as broadened their discussion and improved the overall structure of their manuscript.

Reviewers' comments:

Reviewer's Responses to Questions

**Comments to the Author**

1. If the authors have adequately addressed your comments raised in a previous round of review and you feel that this manuscript is now acceptable for publication, you may indicate that here to bypass the “Comments to the Author” section, enter your conflict of interest statement in the “Confidential to Editor” section, and submit your "Accept" recommendation.

Reviewer #1: All comments have been addressed

Reviewer #2: All comments have been addressed

2. Is the manuscript technically sound, and do the data support the conclusions?

Reviewer #1: Yes

Reviewer #2: Yes

3. Has the statistical analysis been performed appropriately and rigorously? 

Reviewer #1: Yes

Reviewer #2: Yes

4. Have the authors made all data underlying the findings in their manuscript fully available?

Reviewer #1: Yes

Reviewer #2: Yes

5. Is the manuscript presented in an intelligible fashion and written in standard English?

Reviewer #1: Yes

Reviewer #2: Yes

6. Review Comments to the Author

Reviewer #1: I am pleased to report that all the revisions I had previously requested have been satisfactorily addressed. The changes made have significantly enhanced the clarity and depth of the manuscript, ensuring that the scientific merit and reliability of the findings are well represented.

The revisions have adequately resolved the issues raised during the initial review, and I believe that the manuscript is now improved. The discussion is clearer, the methodology is more robust, and the overall presentation aligns well with the journal's standards.

Reviewer #2: The authors have adequately addressed the comments raised in the previous round of review. I found the manuscript improved and now it is suitable for pubblication in this form.

7. PLOS authors have the option to publish the peer review history of their article (what does this mean?). If published, this will include your full peer review and any attached files.

Reviewer #1: No

Reviewer #2: No

---

## [Editor Report · Acceptance letter]

29 Jan 2025

PONE-D-24-28766R1 

PLOS ONE

Dear Dr. Stewart, 

I'm pleased to inform you that your manuscript has been deemed suitable for publication in PLOS ONE. Congratulations! Your manuscript is now being handed over to our production team.

Kind regards, 

on behalf of

Dr. Barry Kweh 

Academic Editor

PLOS ONE